# The Impact of the Inoculation of Different *Pied de Cuve* on the Chemical and Organoleptic Profiles of Wines

**DOI:** 10.3390/microorganisms12081655

**Published:** 2024-08-13

**Authors:** Katherine Bedoya, Albert Mas, Nicolas Rozès, Carla Jara, María del Carmen Portillo

**Affiliations:** 1Biotecnología Enológica, Department Bioquímica i Biotecnologia, Facultat d‘Enologia, Universitat Rovira i Virgili, C/Marcel·lí Domingo 1, 43007 Tarragona, Spain; khaterine.bedoya@urv.cat (K.B.); albert.mas@urv.cat (A.M.); 2Biotecnologia Microbiana dels Aliments, Department Bioquímica i Biotecnologia, Facultat d‘Enologia, Universitat Rovira i Virgili, C/Marcel·lí Domingo 1, 43007 Tarragona, Spain; nicolasrozes@urv.cat; 3Departmento de Agroindustria y Enología, Facultad de Ciencias Agronómicas, Universidad de Chile Santa Rosa, Santiago 11315, Chile; carlajara@u.uchile.cl

**Keywords:** *Pied de cuve*, alcoholic fermentation, wine quality, sensory analysis, HPLC analysis

## Abstract

Controlling the microorganisms involved in alcoholic fermentation during wine production can be achieved by adding a small quantity of spontaneously fermenting must to freshly crushed grapes, a technique known as *pied de cuve* (PdC). This method not only serves as an inoculation starter but also enhances the microbial footprint unique to each wine region. Recent studies have confirmed that wines inoculated with PdC exhibit efficient fermentation kinetics comparable to those inoculated with commercial strains of *Saccharomyces cerevisiae*. However, further research is required to draw robust conclusions about the chemical and sensory impacts of PdC-inoculated wines. In this study, we examined the chemical and sensory effects of the PdC technique across three different harvests: Muscat of Alexandria (Spain, harvests 2022 and 2023) and Sauvignon Blanc (Chile, harvest 2023). Each PdC was prepared using various stressors (sulfur dioxide, ethanol, and temperature). Our findings revealed that wines produced with PdC exhibited similar fermentation kinetics and sensory profiles to those inoculated with commercial strains. Notably, PdC fermentations resulted in lower concentrations of acetic acid compared to both the commercial strain and spontaneous fermentations. The sensory analysis indicated that PdC wines significantly differed from those made with commercial strains, with PdC wines displaying more pronounced tropical notes. These results suggest that the PdC technique, particularly when using specific stressors, can maintain desirable fermentation characteristics while enhancing certain sensory attributes, offering a viable alternative to traditional inoculation methods.

## 1. Introduction

The use of commercial yeast, such as the dry wine yeast (DWY), is a widely adopted inoculation strategy in wineries [1,2]. Its popularity stems from its ability to ensure successful alcoholic fermentation (AF) and achieve desired organoleptic attributes in the final wine, leading to uniformized aromatic profiles [3]. However, DWY usage can be costly, particularly for small and medium-sized cellars, due to the need to purchase additional compounds like nitrogen for adding during the DWY preparation or adding directly into the must to enhance the fermentation performance [1,4,5,6].

Consequently, some winemakers are exploring strategies to integrate native microbiota from the cellar environment into the fermentation process. By this, they are leveraging the adaptability of native yeast strains to specific environmental conditions to produce wines with unique sensory attributes [7,8]. This approach aligns with the environmentally conscious practices of wineries moving towards sustainability and organic methods, responding to the consumer trends for natural and low-intervention wines that reflect the *terroir* concept [9].

Several strategies exist to maintain microbial diversity in winemaking, including spontaneous fermentation, the selection of autochthonous strains, or the *pied of cuve* (PdC) method [8,9]. Spontaneous fermentation (SF) is a process carried out by the non-*Saccharomyces* (non-Sce) yeasts and *Saccharomyces* (Sce) yeasts that are naturally present in grapevine and winery environments [10]. Though SF allows you to conserve the natural microbial diversity, it can lead to off-flavor compounds and sluggish or stuck fermentations. Additionally, it usually results in slower fermentation kinetics compared to must inoculated with the DWY of *Saccharomyces cerevisiae* strains [8]. Another strategy involves selecting yeasts from must or vineyard environments to propagate those with desirable oenological features and preserve and use them for future vintages [11]. Traditionally, some cellars have employed the PdC (bottom of the deposit in French) method, which involves using a small volume of already fermenting must to initiate AF when added to fresh must [8,12]. PdC can be derived mainly from two different strategies: in the first one, a single yeast culture (generally a laboratory-selected strain) is propagated in a suitable medium or must that will be later added as PdC to the main vat of must; the second can be derived from small vats already fermenting spontaneously, either in the cellar or the vineyard. These vats are analyzed for organoleptical and physiochemical properties to select the PdC with the best characteristics and absence of off-flavors for inoculation into a new batch of must [8,12]. While some wineries still use PdC, its impact on yeast populations and wine sensory attributes in wine is not well studied. Some authors highlight the advantages of PdC for microbial control during AF and improved fermentation kinetics compared to SF [12,13]. Nonetheless, debates continue regarding its effectiveness and contribution to wine complexity due to the limited number of studies. Existing research has not shown significant differences in global quality characteristics nor disparities in general regarding chemical parameters like residual sugars, ethanol, and the acidity between the PdC, DWY, or SF [13,14]. Only significant differences were found in the metabolic fingerprint or the volatile composition among the PdC, DWY, and SF [13].

In a previous study conducted by our group [15], we evaluated the impact on yeast populations and fermentation kinetics by using combinations of stressors (sulfur dioxide (SO_2_), ethanol, and temperature) to create four different PdCs. These PdCs were then added to fresh Muscat of Alexandria must to conduct the AF. The PdCs initiated four separate AFs, which were compared to an AF inoculated with a DWY strain of *S. cerevisiae*. Our results showed that all the PdCs served as robust AF starters, leading to higher total yeast populations than fermentations inoculated with the DWY. Depending on the stressors used for PdC preparation, some AFs inoculated with PdC had fermentation kinetics comparable to those inoculated with DWY. Significant differences were observed in the abundance of non-Sce species, particularly at the initial fermentation stages of the AF. Furthermore, the Interdelta analysis of the isolated *S. cerevisiae* strains during AFs inoculated with the PdC and DWY indicated significantly higher diversity indexes in those inoculated with the PdCs compared to those inoculated with the DWY.

In the present study, we hypothesize that PdCs created under stress conditions not only support successful AF and promote the selection of diverse fermentative strains but also potentially result in wines with unique organic compound profiles and favorable organoleptic characteristics. This study will assess the chemical and sensory impact of wines inoculated with PdCs across three harvest campaigns and compare them to those produced through DWY inoculation or SF. The three campaigns were as follows: (i) 2022 harvest using Muscat of Alexandria must in Spain; (ii) 2023 harvest using Sauvignon Blanc must in Chile; and (iii) 2023 harvest using Muscat of Alexandria in Spain. The physicochemical parameters of the wines produced under each treatment will be measured using high-performance liquid chromatography (HPLC), and the organoleptic evaluation will be assessed through a sensory analysis.

## 2. Materials and Methods

### 2.1. Experiment Setting and Sampling

Two of the experiments were conducted at the experimental cellar of the University Rovira I Virgili (Tarragona, Spain), using fresh must from the Muscat of Alexandria variety harvested in the 2022 (M2) and 2023 (M3) campaigns. The third experiment took place at the University of Chile’s cellar (Santiago, Chile), employing fresh must of the Sauvignon Blanc variety harvested in 2023 (S3). The parameters of the musts for the alcoholic fermentation (AF) are described in Table 1.

All musts were settled according to the method described by Bedoya et al. [15] to prepare both the PdC and the must for AF.

The kinetics of PdCs and AFs were measured daily by densitometry using an electronic densitometer (Densito 30PX Portable Density Meter; Mettler Toledo, Barcelona, Spain) until density was under 990 g/L. The total yeast population was monitored using a microscopy counting chamber, and the viable population was tracked by colony-forming unit (CFU) counts on YPD solid medium (1% yeast extract, 2% peptone, 2% dextrose, and 17 g/L Agar, all compounds come from Biogenetics, Milan, Italy); Lysine (LYS, Oxoid Ltd., Basingstoke, UK) and WLN (Difco Laboratories, Detroit, MI, USA).

### 2.2. Pied de Cuve

After settling for 24 h, 1 L of must was placed in 1.5 L bottles for each treatment. Some PdCs were treated with SO_2_ and ethanol (SE), and others were fermented without additives at two temperatures (Table 2). We used the code M2-26 for a PdC from the 2022 Muscat of Alexandria harvest, fermented at 26 °C without additives. The code M2-26SE indicates a PdC from the same harvest and grape variety, fermented at 26 °C with SE additions. The rest of the samples were coded similarly (Table 2). Each PdC treatment was monitored until the density dropped by 15–20 g/L and the microscopy showed approximately 1 × 10^8^ cells/mL. At this point, triplicates of PdCs were combined and added to fresh must at 2% (*v*/*v*) to start AF.

### 2.3. Alcoholic Fermentation

Once the grapes reached maturity with a probable alcohol content of 12% (*v*/*v*), approximately 60 Kg of grapes was harvested and processed to obtain the must. After 24 h of settling as described previously, 3 L of must was distributed into 5 L containers, and the AF was conducted in triplicate at 18 °C without agitation at the cellars. The different inoculation strategies of AFs are described in Table 2. One set of AFs was inoculated with 2% (*v*/*v*) of the PdCs prepared previously and labelled according to the conditions used for the corresponding PdC. The second set of AFs was inoculated with 2 × 10^6^ cells/mL of rehydrated DWY *S. cerevisiae* and 40 mg/L of SO_2_ to serve as control fermentations (labelled as C-). The DWY of *S. cerevisiae* used in the experiments was obtained from Lallemand Inc., Montreal, Canada, differed for the experimentation in Spain and Chile, reflecting the strains commonly used in each cellar. For the experiments M2 and M3 in Spain, the *S. cerevisiae bayanus* strain QA23 was used. During the Chilean campaign S3, the *S. cerevisiae* (ex-*bayanus*) strain 1118 was inoculated. Additionally, for the M3 harvest, we included an AF without any inoculation to conduct spontaneous fermentation (SF-M3).

### 2.4. Chemical Analysis

Samples for the chemical analysis were taken from must, at the beginning and end of each AF. Total sugars and YAN (ammonia and primary amino nitrogen) were quantified using the Y15 Bioanalyzer with the corresponding enzymatic kits (BioSystems S.A, Barcelona, Spain).

The main chemical parameters of the produced wines (glucose, fructose, ethanol, and glycerol) and acids (acetic and succinic acid) were quantified in an Agilent 1100 HPLC (Agilent Technologies, Waldbronn, Germany) following the procedure described by Bedoya et al. [15]. Briefly, the samples were centrifugated at 13,000 rpm for 5 min, and the supernatant was filtered with 0.22 μm pore filters before injection (Agilent Technologies). The main chemical parameters of the wines (glucose, fructose, ethanol, and glycerol) and acids (acetic and succinic acid) were quantified in an Agilent 1100 HPLC (Agilent Technologies, Germany). The HPLC had coupled a Hi-Plex H (300 mm × 7.7 mm) column inside a 1260 MCT (Infinity II Multicolumn Thermostat). The column conditions involved maintaining a temperature of 60 °C for 30 min, with the mobile phase at 5 mM H_2_SO_4_ flowing at a 0.6 mL/min of rate. Additionally, the chromatograph was equipped with two detectors: an MWC detector (G1365B multi-wavelength detector) and a RID detector (1260 Infinity II refractive index detector) (Agilent Technologies, Germany). The results of ethanol are expressed as the yield of grams produced per 100 grams of consumed sugar (g/100 g), averaged from biological triplicates.

The results of ethanol are expressed as the % of efficiency in relation to the theoretical maximum ethanol production per gram of sugar (0.51 g), averaged from biological triplicates.

### 2.5. Sensory Analysis

The wines produced from each experiment were aged in bottles for 2 months at 8 °C before undergoing sensory analysis. Each sensory session involved between 18 and 23 trained and expert tasters in wine that frequently participated in tasting sessions of this type. They came from the Faculty of Oenology (Universitat Rovira I Virgili) for M2 and M3 harvests and from the University of Chile for the S3 harvest. The panelists were informed about the kind of samples they were going to taste and the general conditions of the experiments. The identities of the panelists were protected by assigning random numbers to their answer sheets, corresponding to their place during the analysis. Tasting was carried out with official black glasses to eliminate visual subjectivity related to the color and intensity of wines. Additionally, the glasses were randomly numbered with 3-digit codes, and the wines were served anonymously according to a Latin square of Williams design to avoid the range and carry-over effect.

During these sessions, a triangle test was conducted to assess whether the panelists could distinguish the wines inoculated with the PdCs and their respective controls. For the M2 campaign, comparisons included P-M2-26 vs. C-M2; P-M2-18 vs. C-M2; P-M2-26SE vs. C-M2; and P-M2-18SE vs. C-M2. For the S3 campaign, comparisons included P-S3-26SE vs. C-S3; and P-S3-18SE vs. C-S3. For the M3 campaign, comparisons included P-M3-18SE vs. C-M3; P-M3-18SE vs. SF-M3; and C-M3 vs. SF-M3.

Concurrently, a simple descriptive test was conducted to evaluate attributes such as terpene, tropical, vegetal, acidity, bitterness, and quality. Tasters rated the intensity of each attribute using a structured scale ranging from 1 (no detection) to 5 (highest value). Descriptive results from panelists who were able to differentiate the wines in the triangle test were normalized, resulting in 14 panelists for each descriptive test.

### 2.6. Statistical Analysis

The fermentations were performed in triplicate, and the data were expressed as the mean and standard deviation. The area under the curve (AUC) was used to identify significant differences (*p*-value < 0.05) in fermentation performance, measured by a decrease in must density [16]. Briefly, AUC values were calculated by the integrating density decrease over two consecutive time points. Statistical analysis was performed using ANOVA and the Tukey test with XLSTAT version 2023.2.1413 software (Lumivero, New York, NY, USA).

The same software and the statistical test were used to identify significant differences in organic acids and other compounds across all conditions. The data of the triangle test and the mean of scores of the descriptive attributes during the sensory analysis were analyzed statistically using ANOVA, and the spider plots were created with Excel. The level of significance was 0.05.

## 3. Results and Discussion

### 3.1. Sugar Consumption and Population Dynamics

The yeast population dynamics and sugar consumption kinetics during the PdC prepared under different stressors (SO_2_, ethanol, and temperature) and the AFs inoculated with them were monitored during three harvest campaigns using different grape varieties (M2, S3, and M3). These dynamics are represented in Figure 1, Figure 2 and Figure 3 with their AUC calculated for the AFs in each harvest (Appendix A). The evaluation of stressors used in the PdC preparation helped us to select the parameters for the following harvests. Our criteria included yeast population composition and quantity, must density decrease rate during fermentation, main organic compound production, and the sensory characteristics of the final wines.

#### 3.1.1. M2 Harvest: Antecedents from Our Laboratory

The fermentation kinetics and population dynamics for M2 harvest were recently published by our group [15]. In that study, we assessed the effects of stressors individually and the optimal time to inoculate the PdC into fresh must. We found that 40 mg/L of SO_2_ and 1% (*v*/*v*) ethanol were effective stressors, prolonging the presence of both moderate fermentative non-Sce and highly fermentative Sce. The best inoculation time using natural must was when the PdC experienced a density drop of 10–20 g/L. A lower temperature during the PdC allowed the non-Sce to be more abundant at the beginning of the AF, significantly increasing Sce diversity compared with the control inoculated with a commercial Sce strain. While yeast growth differences during AFs were minimal, total population growth at the end of the AFs was not significantly different from the inoculated control C-M2 (Figure 1).

AUC analysis for AF kinetics revealed differences for the P-M2-26 and P-M2-26SE (Appendix A). In summary, the PdC method using SO_2_ and ethanol addition at specific inoculation times effectively managed fermentation kinetics and yeast dynamics (Figure 1). Additionally, it promoted local yeast diversity and maintained fermentation security and efficiency comparable to commercial yeast strains [15]. However, we did not know if the differences at the population level and diversity of Sce strains had a chemical and sensory impact on the resulting wines or if the organoleptic differences induced by the PdC could be acceptable for the consumers. Herein the present study, we aimed to verify the effectiveness of these selected stressors on PdCs during M2 harvest to conduct the AF in different harvests and grape varieties (M3 and S3). Additionally, we evaluated the chemical and sensory impact of PdC inoculation on wines from the harvests M2, M3, and S3.

#### 3.1.2. S3 and M3 Harvests

For the S3 harvest, PdCs with SO_2_ and ethanol (concentrations selected previously during M2 harvest) were fermented at 26 and 18 °C (Figure 2). Similar to the results obtained during M2 harvest, the temperature was the primary driver of yeast population growth and fermentation kinetics during PdC (Figure 2).

The PdC inoculated into fresh must provided an abundant yeast population, enabling a fast onset of AF comparable to the control (C-S3). However, both P-S3-26SE and C-S3 finished on the 15th day, while the P-S3-18SE finished on the 16th day. The AUC analysis of the density loss over time indicated significant differences for P-S3-26SE kinetics compared to P-S3-18SE and C-S3 (Appendix A). Thus, the AFs inoculated with PdC containing SO_2_ and ethanol and fermented at 26 °C (P-M2-26SE and P-S3-26SE) showed the lowest AUC values (Appendix A), demonstrating consistent fermentation outcomes across M2 and S3 musts inoculated with the same PdC formulation (SO_2_ and ethanol at high temperatures).

For the M3 harvest, we used 18 °C for PdC preparation based on its comparable yeast population growth, the lowest acetic acid yield, and higher positive sensory attributes such as a tropical flavor, as will be shown latter in Section 3.2. The resulting wine (P-M3-18SE) was compared to wines produced with a commercial strain (C-M3) and those undergoing spontaneous fermentation (SF-M3) (Figure 3).

Interestingly, during the M3 harvest, the initial yeast population of the must was lower than in M2 and S3 harvests. This fluctuation could be due to the special climatological conditions of the vintage, as the 2023 vintage was particularly dry in Spain. The initial yeast population in AF-C-M3 (1.5 × 10⁶ CFU/mL) was higher than in P-M3-18SE (3 × 10⁵ CFU/mL) and SF-S3 (1 × 10^3^ CFU/mL). However, P-M3-18SE showed a faster population increase during the beginning of AF, consistent with PdC inoculation during M2 and S3 harvests. SF-M3 had a similar growth rate to P-M3-18SE (Figure 3). P-M3-18SE finished fermentation in 17 days, while C-M3 and SF-M3 finished 2 and 4 days later, respectively. AUCs for density loss kinetics in AFs during M3 harvest (Appendix A) indicated significant differences among the three fermentation modalities, with the fastest kinetics belonging to P-M3-18SE. These results corroborate that the PdC technique provided effective microbiological control during AF, even with a low initial yeast population in natural must.

Building on these results and the previous findings [15], we investigated if these differences could be reflected in variations in the physicochemical attributes and sensory perception of the resulting wines under various inoculation strategies.

### 3.2. Wine Chemical Analysis

The main organic acids, ethanol, glycerol, and residual sugars of the resulting wines were determined by HPLC for each AF triplicate. Organic acids are crucial for wine stability and significantly influence its organoleptic features such as flavor, color, and aroma [17]. We quantified the organic acids focusing on compounds that differed significantly within each harvest (Table 3).

Higher sugar consumption values were consistently observed for AFs inoculated with commercial strains, while lower values correspond to AFs inoculated with PdC prepared at 18 °C. Interestingly, sugar consumption was not significantly different for AFs inoculated with commercial strains and PdC prepared with stressors at higher temperature (P-M2-26SE and P-S3-26SE). Similar results using the PdC of red grape variety must prepared at 26 °C were reported by Moschetti et al. [18]. The higher temperature of the PdC might have selected highly fermentative yeasts like Sce over the non-Sce yeast [19]. Our previous results showed that the non-Sce presence was favored longer at the PdC at the beginning of the AF during M2 harvest when a lower temperature was applied [15]. In the M3 harvest, sugar consumption was similar across the three modalities (P-M3-18SE, C-M3, and SF-M3).

No significant differences were detected in the efficiency of ethanol production among treatments within each harvest harvests. AFs inoculated with PdCs prepared with SO_2_ and ethanol at 18 °C presented the lowest glycerol concentration for the M2 and M3 harvests, while the influence of the PdC over the glycerol production during S3 harvest was not as prominent. The glycerol concentration in harvest from the same grape variety, M2 and M3, ranged from 3.10 to 6.28 g/L, which falls within the normal range of 2 to 10 g/L [17,20,21,22]. A significant increase in viscosity could be observed for values around 10 g/L [23,24]. Our results indicate that the observed variations could hardly influence the sensory experience of our wines. Glycerol production is influenced by factors such as grape variety, yeast strains, fermentation parameters like temperature, and winemaking processes [20,24,25]. Ruiz-de-Villa et al. [16] have reported values of glycerol for Muscat of Alexandria between 4.86 and 5.97 g/L in musts inoculated with *T*. *delbrueckii* or *S*. *cerevisiae* at different temperatures (25 and 16 °C). They observed an increase of glycerol concentration in wines inoculated with *T. delbrueckii* at 25 °C. Thus, glycerol production is temperature dependent with higher values at higher temperatures [20]. In our study, the effect of PdC prepared with SO_2_ and ethanol and fermented at low temperatures during M2 and M3 harvest might have limited the growth of glycerol-producing yeast [26]. In the wines from S3 harvest, concentrations were between 5.17 and 5.29 g/L, which falls within the previously reported values for wines from the Sauvignon Blanc variety between 5.42 and 6.31 g/L [22].

Succinic acid concentrations were similar among treatments in M2. Significant differences were found just in the M3 harvest, where the C-M3 had the highest value and the SF-M3 had the lowest. Thus, the effect of PdC was not prominent at the succinic acid levels.

A high concentration of acetic acid in wine is not desirable because of the generation of off-flavors and consequent organoleptic depreciation. SF is frequently associated with high volatile acidity due to the proliferation of spoilage yeasts [27]. Acetic acid yields showed no significant differences between the different treatments of M2 harvest. Interestingly, for the S3 and M3 harvests, the lowest acetic yield corresponded to the AFs inoculated with PdCs prepared at 18 °C with the same stress conditions (P-S3-18SE and P-M3-18SE). These results suggest that the combination of low temperature and the stressors in the PdC selected native non-Sce and Sce populations with a low production of acetic acid. In general, acetic acid values obtained during our PdC-inoculated fermentations were between 0.34 and 0.48 g/L, which is similar to the range of 0.33 to 0.49 g/L reported previously [12,13]. The exception was the AF-S3-26SE (0.70 g/L of acetic acid). This result indicates that the PdC elaborated at 18 °C efficiently kept the acetic acid lower than their respective controls and PdC elaborated at a higher temperature.

Further insights into sensory perception were pursued through the triangle and descriptive tests conducted on the final wines.

### 3.3. Wine Sensory Analysis

Sensory analysis was conducted on wines inoculated with different PdCs and their corresponding controls within each harvest (M2, S3, and M3). A triangle test was used to assess if panelists were able to distinguish wines inoculated with the PdCs from their respective controls. Additionally, experts were asked to do a descriptive test of the wines. We employed the same attribute and score scale for sensory evaluation in all harvests (M2, S3, and M3). However, our goal was to assess these attributes within each harvest rather than compare them across the three harvests. It is essential to consider the differences arising from various cultural and personal factors when conducting a sensory analysis of wine on an international scale. The preferences and perceptions of wine quality and factors as tropical can vary greatly between nationalities, limiting definitive conclusions. To achieve a comprehensive and balanced understanding, it is beneficial to include individuals from diverse nationalities and backgrounds in tasting panels and to account for these cultural factors when interpreting the results. The sensory analysis aimed to determine if the PdC-inoculated wines resulted in a consistent variation of the sensory perception and if their overall quality and organoleptic evaluation were positive.

Table 4 summarizes the wine comparisons and the statistical results of the triangular test.

The results of the descriptive test of wines from each harvest were represented in a spiderweb diagram (Figure 4). In the M2 harvest, the wines from the C-M2 presented higher acidity and vegetal and bitterness attributes, which were the main factors differentiating them from the PdC-inoculated wines, both at the descriptive analysis and triangular test. However, as presented in Section 3.2, the wines of the M2 harvest did not show significant differences in acetic acid or ethanol yields, which could have contributed to these attributes. A combination of higher sugar consumption and lower glycerol observed for C-M2 wines could make their taste more tart or sour, explaining the differences in sensory description [28]. In contrast, the P-M2-18SE wines showed high scores for tropical and quality attributes. Terpene, a characteristic feature of the Muscat of Alexandria variety, scored higher in PdC-inoculated wines than C-M2 (Figure 4A). In the M3 harvest, elaborated with the same grape variety, the wines from P-M3-18SE showed a higher score in terpene than SF-M3 but lower than C-M3 (Figure 4C). The tropical, acidity, and quality attributes of P-M3-18SE were similar among the treatments during the M3 harvest. The wines resulting from SF-M3 presented the highest vegetal attribute (Figure 4C), which is generally perceived as a negative one. The wines from the S3 harvest produced by the inoculation of PdCs were also clearly different from the DWY-inoculated control (C-S3) by its higher tropical features in the case of wines from P-S3-18SE (Figure 4B). The lowest quality and terpene were found for P-S3-26SE wines, indicating that the higher temperature during the PdC preparation had a negative influence on the organoleptic perception, similar to the M2 harvest wines. The rest of the attributes of PdC-inoculated wines in S3 harvest were comparable to the C-S3, but the mentioned differences were sufficient for the panelists to differentiate them.

Our results generally agree with [13], who found differences in some attributes but not in the global variable that included quality when comparing PdC-inoculated wines with a DWY-inoculated control and SF. Nonetheless, it should be noted that the perception of specific concepts, such as tropical and quality, varies among tasters from different countries due to geographical and gastronomic cultural differences, and this can limit definitive conclusions.

Overall in our study, the sensory analysis confirmed that PdC inoculation, particularly with stressors like SO₂ and ethanol, influenced the sensory attributes of wines, enhancing specific desirable flavors and aromas while maintaining acceptable levels of volatile acidity and glycerol content.

Future studies should explore the dynamics of Sce strains in must inoculated with PdC. It is crucial to extend this research to other grape varieties, including more complex wines such as red and rosé, to gather additional data that could be applied at the cellar level.

## 4. Conclusions

Our findings have confirmed that the application of selected stressors over PdC can effectively facilitate the AF process. We observed consistent results in yeast population growth and fermentation kinetics across the grape varieties Muscat of Alexandria and Sauvignon Blanc, harvested over three different vintages. These results were comparable to those obtained by the DWY-inoculated controls. Additionally, the stressors applied during the preparation of the PdC, especially temperature, significantly influenced the performance of the PdC and the selection of yeast strains driving the different AFs. This variability in yeast selection and AF performance highlights the critical role of PdC conditions in shaping fermentation outcomes. Wines produced from PdC-inoculated fermentations exhibited significant differences in certain chemical parameters compared to those from spontaneous and DWY-inoculated fermentations. These differences were also perceptible at the sensory level. Notably, the quality scores of the PdC-inoculated wines were comparable or higher than those of the respective controls at each harvest. Thus, the use of PdC with selected stressors ensured a consistent AF process across different vintages, demonstrating the robustness of this methodology in maintaining desirable fermentation characteristics and wine quality. Future research should be directed to test the effectiveness of the PdC methodology with the selected stressors in other grape varieties and more complex wines. Also, the evaluation of the effect of parameters such as the grape ripeness or viticultural practices on the initial microbiology of the must and the selection of yeasts during PdC is key knowledge to test the robustness of our proposed methodology to control the AF while preserving the autochthonous microbiota.

## Figures and Tables

**Figure 1 microorganisms-12-01655-f001:**
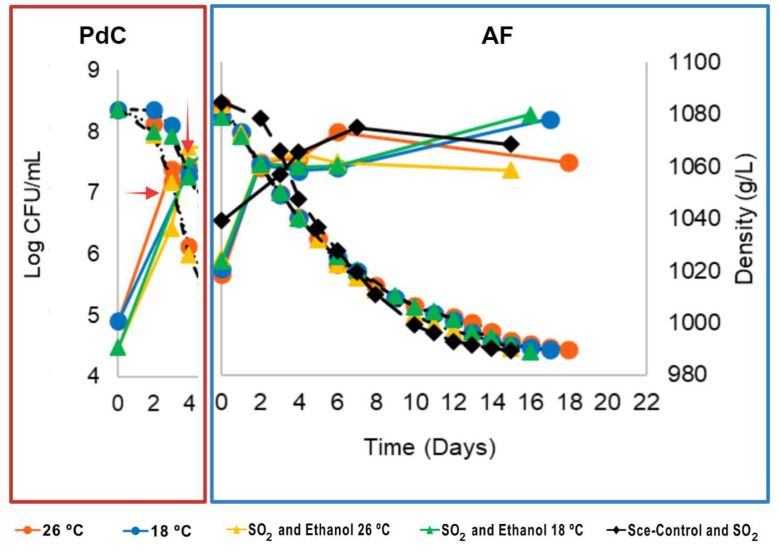
PdC and AF monitoring during M2 harvest, Muscat of Alexandria 2022. Yeast population measured as CFU/mL (solid lines), and fermentation kinetics (dashed lines) represented as must density decrease during *pied de cuve* (PdC) (red rectangle) and the respective alcoholic fermentation (AF) (blue rectangle). Different symbols represent various PdC treatments, and the same color scheme represents the AFs inoculated with their corresponding PdC. Black rhombuses represent control fermentations inoculated with a commercial strain of *S. cerevisiae*. The red arrow indicates the inoculation point of the PdC into fresh must.

**Figure 2 microorganisms-12-01655-f002:**
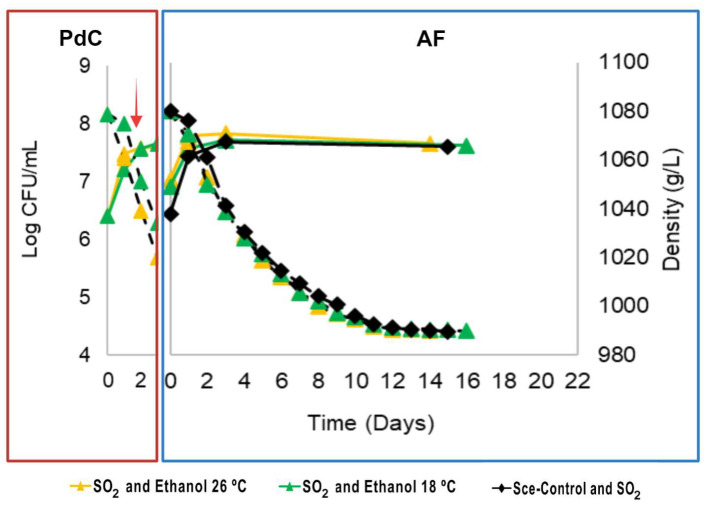
PdC and AF monitoring during S3 harvest, Sauvignon Blanc 2023. Yeast population measured as CFU/mL (solid lines), and fermentation kinetics (dashed lines) represented as must density decrease during *pied de cuve* (PdC) (red rectangle) and the respective alcoholic fermentation (AF) (blue rectangle). Different symbols represent various PdC treatments, and the same color scheme represents the AFs inoculated with their corresponding PdC. Black rhombuses represent fermentation controls inoculated with a commercial strain of *S. cerevisiae*. The red arrow indicates the inoculation point of the PdC into fresh must.

**Figure 3 microorganisms-12-01655-f003:**
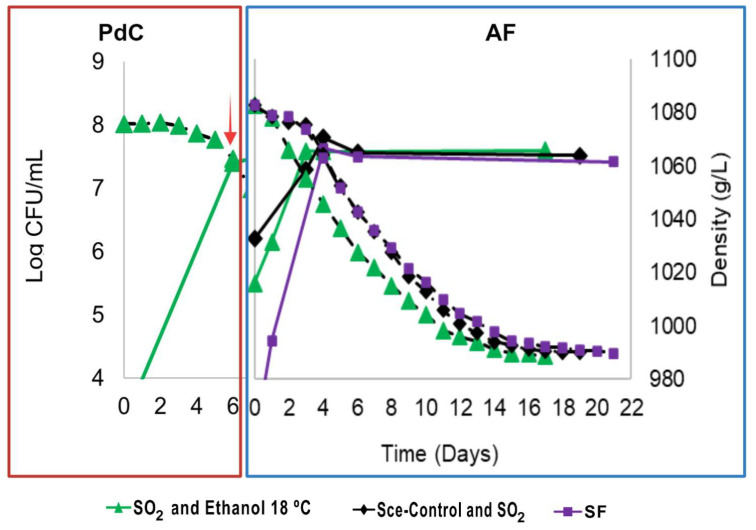
AF monitoring during M3 harvest, Muscat of Alexandria 2023. Yeast population measured as CFU/mL (solid lines), and fermentation kinetics (dashed lines) represented as must density decrease during *pied de cuve* (PdC) (red rectangle) and the respective alcoholic fermentation (AF) (blue rectangle). Different symbols represent various PdC treatments, and the same color scheme represents the AFs inoculated with their corresponding PdC. Black rhombuses represent fermentation controls inoculated with a commercial strain of *S. cerevisiae*, while purple squares indicate musts fermented spontaneously. The red arrow indicates the inoculation point of the PdC into fresh must.

**Figure 4 microorganisms-12-01655-f004:**
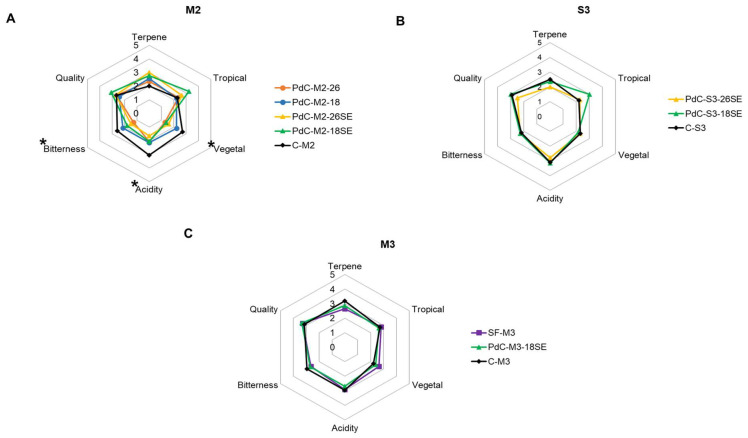
Spiderweb plots for the descriptors terpene, tropical, acidity, vegetal, bitterness, and quality of wines from harvest. (**A**) M2, Muscat of Alexandria 2022; (**B**) S3, Sauvignon Blanc 2023; and (**C**) M3, Muscat of Alexandria 2023. Assessors for each sensory session: 14. Asterisks (*) indicate significant differences between treatments with *p*-value < 0.05.

**Table 1 microorganisms-12-01655-t001:** Main experimental parameters of musts during 2022 and 2023 campaigns.

Harvest	Variety	Density (g/L)	Sugar (g/L)	YAN (mg/L)	pH
M2	Muscat of Alexandria	1082.1	199.8	102.5	3.5
S3	Sauvignon Blanc	1080.0	188.0	98.0	3.3
M3	Muscat of Alexandria	1082.6	217.6	82.0	3.4

**Table 2 microorganisms-12-01655-t002:** *Pied de cuve* (PdC) treatments and alcoholic fermentation (AF) of musts inoculated with different PdC. The M2 and M3 harvest correspond to Muscat of Alexandria must from vintages 2022 and 2023 at Tarragona. The harvest S3 correspond to Sauvignon Blanc must collected at the 2023 campaign at Santiago de Chile.

Harvest	Spontaneous Fermentation for PdC		AF at 18 °C and No Stress	
	Code	Parameters	Code	Inoculation Modality
M2	M2-26	26 °C and no stress	P-M2-26	Inoculated with M2-26
	M2-18	18 °C and no stress	P-M2-18	Inoculated with M2-18
	M2-26SE	26 °C with SO_2_ and EtOH	P-M2-26SE	Inoculated with M2-26SE
	M2-18SE	18 °C with SO_2_ and EtOH	P-M2-18SE	Inoculated with M2-18SE
	Null		C-M2	Control inoculated with Sce QA23
S3	S3-26SE	26 °C with SO_2_ and EtOH	P-S3-26SE	Inoculated with S3-26SE
	S3-18SE	18 °C with SO_2_ and EtOH	P-S3-18SE	Inoculated with S3-18SE
	Null		C-S3	Inoculated with Sce 1118
M3	M3-18SE	18 °C with SO_2_ and EtOH	P-M3-18SE	Inoculated with M3-18SE
	Null		C-M3	Control inoculated with Sce QA23
	Null		SF-M3	Spontaneous fermentation

**Table 3 microorganisms-12-01655-t003:** Chemical analysis of the final wines from each AF modality. M2, Muscat of Alexandria 2022; M3, Muscat of Alexandria 2023; and S3, Sauvignon Blanc 2023. All data are expressed as the arithmetic average of three biological replicates ± standard deviation (n = 3). The % of theoretical yield for ethanol production is included (maximum of 0.51 g ethanol/g consumed sugar). The added ethanol to PdC corresponds to 0.02% over the final quantity.

Harvest	AF Modality	Sugar Consumption (g/L)	Ethanol (g/L)	% Theoretical Ethanol Yield	Glycerol (g/L)	Succinic Acid (g/L)	Acetic Acid (g/L)
M2	P-M2-26	186.79 ± 0.69 ^a^	89.5 ± 6.2	94.0 ± 6.8	4.41 ± 0.02 ^a^	1.04 ± 0.10 ^a^	0.35 ± 0.04 ^a^
	P-M2-18	176.61 ± 0.13 ^b^	82.0 ± 7.3	91.1 ± 8.1	3.90 ± 0.19 ^ab^	1.18 ± 0.03 ^a^	0.35 ± 0.01 ^a^
	P-M2-26SE	182.85 ± 2.53 ^c^	89.6 ± 8.0	96.1 ± 8.9	4.05 ± 0.10 ^ab^	0.99 ± 0.06 ^a^	0.35 ± 0.02 ^a^
	P-M2-18SE	175.39 ± 0.49 ^b^	88.6 ± 5.2	99.1 ± 6.1	3.10 ± 0.17 ^c^	1.00 ± 0.03 ^a^	0.34 ± 0.04 ^a^
	C-M2	189.35 ± 0.46 ^a^	83.8 ± 8.9	86.8 ± 9.3	3.54 ± 0.24 ^bc^	1.06 ± 0.01 ^a^	0.39 ± 0.01 ^a^
S3	P-S3-26SE	178.91 ± 0.92 ^ab^	88.7 ± 1.8	97.0 ± 1.9	5.29 ± 0.05 ^a^	N.D.	0.70 ± 0.03 ^a^
	P-S3-18SE	177.19 ± 0.92 ^b^	87.4 ± 0.6	96.5 ± 1.4	5.29 ± 0.04 ^a^	N.D.	0.48 ± 0.01 ^b^
	C-S3	181.50 ± 0.35 ^a^	88.9 ± 2.8	95.8 ± 3.1	5.17 ± 0.04 ^b^	N.D.	0.76 ± 0.04 ^a^
M3	P-M3-18SE	216.13 ± 0.06 ^a^	95.6 ± 0.3	86.7 ± 0.2	5.10 ± 0.03 ^a^	1.28 ± 0.03 ^ab^	0.46 ± 0.03 ^a^
	C-M3	215.33 ± 0.02 ^a^	94.8 ± 0.7	86.7 ± 0.6	6.32 ± 0.03 ^b^	1.39 ± 0.01 ^b^	0.69 ± 0.01 ^b^
	SF-M3	215.83 ± 1.03 ^a^	95.4 ± 0.8	86.7 ± 1.2	6.28 ± 0.08 ^b^	1.16 ± 0.04 ^a^	0.86 ± 0.01 ^c^

N.D. means not determined. Different letters indicate that the values are significantly different within the same harvest (*p*-value < 0.05).

**Table 4 microorganisms-12-01655-t004:** Results of triangular sensory analysis of wines obtained according to the type of inoculation of natural musts. M2, Muscat of Alexandria 2022; M3, Muscat of Alexandria 2023; and S3, Sauvignon Blanc 2023. Assessors for each sensory session: 14.

Harvest	Comparison	*p*-Value
M2	P-M2-26 vs. C-M2	**<0.05**
	P-M2-18 vs. C-M2	**<0.05**
	P-M2-26SE vs. C-M2	0.175
	P-M2-18SE vs. C-M2	**<0.05**
S3	P-S3-26SE vs. C-S3	**<0.05**
	P-S3-18SE vs. C-S3	**<0.05**
M3	P-M3-18SE vs. SF-M3	0.179
	P-M3-18SE vs. C-M3	**<0.05**
	SF-M3 vs. C-M3	**<0.05**

Significantly different values (*p*-value < 0.05) are highlighted in bold.

## Data Availability

The original contributions presented in the study are included in the article/Appendix A, further inquiries can be directed to the corresponding author.

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
