# Peer review of "The Impact of the Inoculation of Different *Pied de Cuve* on the Chemical and Organoleptic Profiles of Wines"

_microorganisms, 2024, doi:10.3390/microorganisms12081655_

Round 1
Reviewer 1 Report
Comments and Suggestions for Authors
The manuscript 'Impact of the inoculation of different pied de cuve on the chemical and organoleptic profile of wines' presents quite interesting results which may be usefull in the production of wine with unique sensory profile. Before accepting for publication in Microorganisms journal the manuscript needs some improvements, as given below:
Introduction - line 34 Dry Wine Yeast (DWY) - in the reviewer's opinion, it should be written in lowercase letters (i.e. dry wine yeast), even though the abbreviation is in uppercase letters
2.1. Experiments setting and sampling
line 111 - 'The kinetics of PdCs and AFs were measured daily by densitometry.' please describe in more detail
2.2. Pied de cuve
Line 125 'x 108 cells/mL' - Is this the total number of yeast cells? that's unfortunate that there was no in-depth analysis of Pied de cuve, e.g. what species of microorganisms and in what quantity are present.
2.3. Alcoholic fermentation
lines 139-140 - The authors wrote: 'The second set of AFs was inoculated with 2 x 106 cells/mL of rehydrated DWY S. cerevisiae and 40 mg/L of SO2, to serve as control fermentations (labelled as C-).' Was ethanol added to these samples or not? If so, at what concentration?
2.4. Chemical analysis
It is suggested to give in more details conditions of HPLC analysis
2.5. Sensory analysis - more details on panellists is required. Was the study performed in compliance with GDPR? Were the study participants clearly informed about the test and possible risks and had the opportunity to ask questions before they gave their consent?
Table 3 - ethanol yield - it would be better to supplement the table with the unit. Moreover, it is recommended to give ethanol production efficiency in % of theoretical
line 312 - T. delbrueckii or S. cerevisiae - should be written in italics
Author Response
Reviewer#1
The manuscript 'Impact of the inoculation of different pied de cuve on the chemical and organoleptic profile of wines' presents quite interesting results which may be usefull in the production of wine with unique sensory profile. Before accepting for publication in Microorganisms journal the manuscript needs some improvements, as given below:
Introduction - line 34 Dry Wine Yeast (DWY) - in the reviewer's opinion, it should be written in lowercase letters (i.e. dry wine yeast), even though the abbreviation is in uppercase letters
Change as suggested (line 34).
2.1. Experiments setting and sampling
line 111 - 'The kinetics of PdCs and AFs were measured daily by densitometry.' please describe in more detail
We have included more detail about densitometry measurements in lines 111-113 and now it can be read “The kinetics of PdCs and AFs were measured daily by densitometry using an electronic densitometer (Densito 30PX Portable Density Meter; Mettler Toledo, Barcelona, Spain) until density was under 990 g/L”.
2.2. Pied de cuve
Line 125 'x 108 cells/mL' - Is this the total number of yeast cells? that's unfortunate that there was no in-depth analysis of Pied de cuve, e.g. what species of microorganisms and in what quantity are present.
Yes, the number correspond to the total yeast cells counted under the microscope. This was the faster method to determine when the PdC harbored the appropriated population to be inoculated into the new must to start the AF. The objective of the present study was to corroborate that PdC was effective to successfully finish the alcoholic fermentation during wine production of different grape varieties and harvests while assessing the chemical and sensory impacts of PdC-inoculated wines. Nonetheless, the recent published article of our group “Bedoya et al. 2024” investigated the effect of the different parameters over the selection of yeast during PdC and the subsequent AF of M2 samples, i.e, Muscat of Alexandria from 2022 harvest. In Bedoya et al. (2024) CFU/mL and qPCR were employed to follow yeast dynamics during PdC and AF.
Bedoya, K., Buetas, L., Rozès, N., Mas, A. & Portillo, M. C. Influence of different stress factors during the elaboration of grape must’s pied de cuve on the dynamics of yeast populations during alcoholic fermentation. Food Microbiology 123, (2024).
2.3. Alcoholic fermentation
lines 139-140 - The authors wrote: 'The second set of AFs was inoculated with 2 x 106 cells/mL of rehydrated DWY S. cerevisiae and 40 mg/L of SO2, to serve as control fermentations (labelled as C-).' Was ethanol added to these samples or not? If so, at what concentration?
Ethanol was not added to control fermentations. The inoculation with DWY followed the standard procedure and routine of the cellar (2 x 106 cells/mL of rehydrated DWY S. cerevisiae and 40 mg/L of SO2).
2.4. Chemical analysis
It is suggested to give in more details conditions of HPLC analysis
Thank you for the suggestion. To included more details of HPLC analysis, we have complemented the reference Bedoya et al. [15] by the following paragraph in lines 155-167:
“Briefly, samples were centrifugated at 13000 rpm for 5 min and the supernatant was filtered with 0.22 μm pore filters before injection (Agilent Technologies). The main chemical parameters of the wines (glucose, fructose, ethanol, glycerol) and acids (acetic and succinic acid) were quantified in an Agilent 1100 HPLC (Agilent Technologies, Germany). The HPLC had coupled a Hi-Plex H (300 mm × 7.7 mm) column inside a 1260 MCT (Infinity II Multicolumn Thermostat). The column conditions involved maintaining a temperature of 60 ºC for 30 min, with the mobile phase at 5 mM H2SO4 flowing at a 0.6 mL/min of rate. Additionally, the chromatograph was equipped with two detectors: an MWC detector (G1365B multi-wavelength detector) and a RID detector (1260 Infinity II refractive index detector) (Agilent Technologies, Germany).”
2.5. Sensory analysis - more details on panellists is required. Was the study performed in compliance with GDPR? Were the study participants clearly informed about the test and possible risks and had the opportunity to ask questions before they gave their consent?
Thanks for the suggestion. All the sensory analysis performed for research at the Faculty of Oenology (Tarragona) are performed by experts tasters in wine, they are anonymous and informed about the kind of samples they are going to analyse through the general call for participation. Additionally, during tasting they can ask general and specific doubts to the organizers.
In the current version we have completed the sensory analysis information (lines 174-183): “Each sensory session involved between 18 and 23 trained and expert tasters in wine that frequently participated in tasting sessions of this type. They came from the Faculty of Oenology (Universitat Rovira I Virgili) for M2 and M3 harvests, and from the University of Chile for the S3 harvest. Panellists were informed about the kind of samples they were going to taste and the general conditions of the experiments. The identities of the panellists were protected by assigning random numbers to their answer sheets, corresponding to their place during the analysis. Tasting was carried out with the official black glasses to eliminate visual subjectivity related to the colour and intensity of wines. Additionally, glasses were randomly numbered with 3-digit codes and wines were served anonymously according to a Latin square of Williams design to avoid the range and carry-over effect.”
Table 3 - ethanol yield - it would be better to supplement the table with the unit. Moreover, it is recommended to give ethanol production efficiency in % of theoretical
We have substituted in table 3 the ethanol yield (g ethanol/100g of consumed sugar) by the % of theoretical ethanol production (maximum 0,51g ethanol/g sugar), as suggested. Additionally, we have modified the material and methods section (line 168-170) together with results and discussion section (line 324) to include the ethanol efficiency instead of ethanol yield.
line 312 - T. delbrueckii or S. cerevisiae - should be written in italics
Thank you for noticing the error. It has been corrected (line 335-336).
Reviewer 2 Report
Comments and Suggestions for Authors
Interesting work exploring the role of PdC additions and their impact on different wine parameters. Overall, the manuscript is clear and well structured.
Having harvest from different countries and grape cultivars might be confusing to the reader, but it is explained in the manuscript.
There are just a few suggestions and comments to address to the sensory data.
Were the 14 assessors used for M2 and M3 the same? Experts were winemakers, experts in oenology, food science?
Add some short information about how the sensory evaluation procedure was done. Blind? In black glasses? Randomised and coded?
Were ANOVA analysis performed on the sensory data? Or at least, the std error for each attribute that would reflect the variability within assessors.
This information will add value to the paragraph. Otherwise, having only the spider plots might not be enough to state that certain attributes were perceived higher than others.
Also, the authors clearly describe that comparison across harvest cannot be made as wines are made of different cultivars and countries. Nonetheless, my concern is more about the use of terms - the perception of specific concepts (i.e., tropical, and quality) might not be the same between both countries. How can it be shown that assessors in Spain and Chile have the exact same associations to tropical fruits?
How can it be shown that assessors in both countries had the same perception of wine quality?
If might not be possible to show that in this study. However, even if there is not a comparison between vintages and cultivars, it will be worth to address as potential limitation this in the text.
Author Response
Reviwer2
Interesting work exploring the role of PdC additions and their impact on different wine parameters. Overall, the manuscript is clear and well structured.
Having harvest from different countries and grape cultivars might be confusing to the reader, but it is explained in the manuscript.
There are just a few suggestions and comments to address to the sensory data.
Were the 14 assessors used for M2 and M3 the same? Experts were winemakers, experts in oenology, food science?
Following both reviewer’s suggestions we have completed the sensory analysis information (lines 174-183): “Each sensory session involved between 18 and 23 trained and expert tasters in wine that frequently participated in tasting sessions of this type. They came from the Faculty of Oenology (Universitat Rovira I Virgili) for M2 and M3 harvests, and from the University of Chile for the S3 harvest. Panellists were informed about the kind of samples they were going to taste and the general conditions of the experiments. The identities of the panellists were protected by assigning random numbers to their answer sheets, corresponding to their place during the analysis. Tasting was carried out with official black glasses to eliminate visual subjectivity related to the colour and intensity of wines. Additionally, glasses were randomly numbered with 3-digit codes and wines were served anonymously according to a Latin square of Williams design to avoid the range and carry-over effect.”
Add some short information about how the sensory evaluation procedure was done. Blind? In black glasses? Randomised and coded?
Please, see previous answer.
Were ANOVA analysis performed on the sensory data? Or at least, the std error for each attribute that would reflect the variability within assessors.
Yes, ANOVA analysis was also performed on the sensory data. We have added this information in the section 2.6 Statistical analysis and now, in lanes 203-206, can be read: “The data of the triangle test and the mean of scores of the descriptive attributes during the sensory analysis were analysed statistically using ANOVA and the spider plots were created with Excel. The level of significance was 0.05.”
This information will add value to the paragraph. Otherwise, having only the spider plots might not be enough to state that certain attributes were perceived higher than others.
We agree with the reviewer and now we have added more detail in the lines 210-213, as commented in the previous answer. Additionally, asterisks were included on the spider plots of Figure 4 to indicate significant differences in the attribute’s perception, modifying accordingly the description of Figure 4.
Also, the authors clearly describe that comparison across harvest cannot be made as wines are made of different cultivars and countries. Nonetheless, my concern is more about the use of terms - the perception of specific concepts (i.e., tropical, and quality) might not be the same between both countries. How can it be shown that assessors in Spain and Chile have the exact same associations to tropical fruits? How can it be shown that assessors in both countries had the same perception of wine quality? If might not be possible to show that in this study. However, even if there is not a comparison between vintages and cultivars, it will be worth to address as potential limitation this in the text.
Thank you for the suggestion, we totally agree with your comment. We have now added this limitation and consideration in lines 370-376: “It is essential to consider the differences arising from various cultural and personal factors when conducting sensory analysis of wine on an international scale. The preferences and perceptions of wine quality and factors as tropical can vary greatly between nationalities limiting definitive conclusions. To achieve a comprehensive and balanced understanding, it's beneficial to include individuals from diverse nationalities and backgrounds in tasting panels and to account for these cultural factors when interpreting the results.”
Round 2
Reviewer 1 Report
Comments and Suggestions for Authors The manuscript has been revised in accordance with the reviewer's recommendations. However, it would also be advisable to provide the ethanol concentration in g/L in Table 3. Moreover, the 100% ethanol yield for the S3 set is surprising. Please explain this necessary. The authors wrote: "No significant differences were detected in the efficiency of ethanol production among treatments within M2 S3, and M3 harvests." Are there really no statistically significant differences in ethanol yields between above mentioned harvests?Author Response
The manuscript has been revised in accordance with the reviewer's recommendations. However, it would also be advisable to provide the ethanol concentration in g/L in Table 3.
We have included the g/L of ethanol in table 3 in addition to the % of the maximum ethanol production, as suggested by reviewer 1.
Moreover, the 100% ethanol yield for the S3 set is surprising. Please explain this necessary.
If the sugar content is underestimated (for example, not accounting for all fermentable sugars), the calculated theoretical yield (TEY) would be lower, potentially giving an efficiency above 100%. We have included this possibility in the revised version of the manuscript (Ln 325-327).
The authors wrote: "No significant differences were detected in the efficiency of ethanol production among treatments within M2 S3, and M3 harvests." Are there really no statistically significant differences in ethanol yields between above mentioned harvests?
As stated in the text "no significant differences were detected in the efficiency of ethanol production among treatments within M2, S3, and M3 harvests", This means that we have compared the ethanol values for each harvest individually, not for all the harvest together". As it can be observed, within each harvest values are highly similar. All the statistical comparisons have been performed within each harvest because we wanted to evaluate the treatment but not the grape variety or harvest. To avoid misleading information we have modified slightly the text and now can be read "No significant differences were detected in the efficiency of ethanol production among treatments within each harvest"